# The Interaction between SARS-CoV-2 Nucleocapsid Protein and UBC9 Inhibits MAVS Ubiquitination by Enhancing Its SUMOylation

**DOI:** 10.3390/v15122304

**Published:** 2023-11-24

**Authors:** Congcong Huang, Yiping Yin, Pan Pan, Yanping Huang, Siwei Chen, Junkai Chen, Ju Wang, Guoqing Xu, Xuan Tao, Xiao Xiao, Jian Li, Jing Yang, Zhixiong Jin, Bei Li, Zhaohui Tong, Weixing Du, Long Liu, Zhixin Liu

**Affiliations:** 1Department of Infectious Diseases, Renmin Hospital, School of Basic Medical Sciences, Hubei University of Medicine, Shiyan 442000, Chinajuwang0809@163.com (J.W.); yangjing780228@foxmail.com (J.Y.);; 2Institute of Virology, Hubei University of Medicine, Shiyan 442000, China; 3Hubei Key Laboratory of Embryonic Stem Cell Research, Hubei University of Medicine, Shiyan 442000, China; 4Department of Respiratory and Critical Care Medicine, Beijing Institute of Respiratory Medicine and Beijing Chao-Yang Hospital, Capital Medical University, Beijing 100054, China; 5The First Affiliated Hospital of Jinan University, Guangzhou 510632, China

**Keywords:** SARS-CoV-2, nucleocapsid protein, UBC9, MAVS, SUMOylation, ubiquitination

## Abstract

Severe COVID-19 patients exhibit impaired IFN-I response due to decreased IFN-β production, allowing persistent viral load and exacerbated inflammation. While the SARS-CoV-2 nucleocapsid (N) protein has been implicated in inhibiting innate immunity by interfering with IFN-β signaling, the specific underlying mechanism still needs further investigation for a comprehensive understanding. This study reveals that the SARS-CoV-2 N protein enhances interaction between the human SUMO-conjugating enzyme UBC9 and MAVS. Increased MAVS-UBC9 interaction leads to enhanced SUMOylation of MAVS, inhibiting its ubiquitination, resulting in the inhibition of phosphorylation events involving IKKα, TBK1, and IRF3, thus disrupting IFN-β signaling. This study highlights the role of the N protein of SARS-CoV-2 in modulating the innate immune response by affecting the MAVS SUMOylation and ubiquitination processes, leading to inhibition of the IFN-β signaling pathway. These findings shed light on the complex mechanisms utilized by SARS-CoV-2 to manipulate the host’s antiviral defenses and provide potential insights for developing targeted therapeutic strategies against severe COVID-19.

## 1. Introduction

During infection with RNA viruses, such as SARS-COV-2, pathogen-associated molecular patterns (PAMPs) are recognized by retinoic acid-inducible gene I (RIG-I)-like receptors (RLRs) [1]. Following the recognition of PAMPs, specifically double-stranded viral RNA as well as 5′-PPP viral RNA, RIG-I undergoes dephosphorylation and subsequent polyubiquitination mediated by the ubiquitin E3 ligase tripartite-motif protein 25 (TRIM25) or Riplet (also known as REUL or RNF135) [2]. Ubiquitinated RIG-I then interacts with the mitochondrial antiviral signaling protein (MAVS). Upon RIG-I interaction, MAVS is ubiquitinated by TRIM31, leading to the activation of TANK-binding kinase 1 (TBK1) and inhibitor of κB kinase-ε (IKKε), resulting in the phosphorylation of interferon regulatory factor 3 (IRF3) and the production of interferon β (IFN-β). Ultimately, IFN-β binds to specific receptors, inducing the expression of interferon-stimulated genes (ISGs) and initiating a host’s antiviral response [3]. In this process, RIG-I activates MAVS and initiates the first line of defense against viral replication.

Severe and critical COVID-19 patients exhibit a distinct phenotype characterized by a severe impairment of the type I interferon (IFN I) response. This is evident from the absence of IFN-β production and reduced IFN-α production and activity, leading to a persistent viral load in the bloodstream and an exacerbated inflammatory response [4]. SARS-CoV-2 has been shown to encode multiple proteins that manipulate or evade the host’s antiviral response, enabling successful infection [5,6,7]. The nucleocapsid protein (N), the most conservative protein among human β coronaviruses, including SARS-CoV-2, responsible for COVID-19, has been implicated in interfering with the host’s innate immune response. The SARS-COV-2 N protein has been previously documented to interact with RIG-I at the DExD/H domain, inhibiting type I interferon (IFN I) signaling by disrupting the binding of immunostimulatory RNAs [8].

A recent study indicates that the N protein of SARS-CoV-2 may employ additional mechanisms to evade the host’s innate immune response. One such mechanism involves the interaction between the N protein and the MAVS, hindering the IFN-β signaling pathway by suppressing the Lys63-linked poly-ubiquitination of MAVS [9]. However, the precise underlying mechanism still requires further investigation for a comprehensive understanding.

In this study, we revealed that the N protein plays a crucial role in modulating MAVS SUMOylation by promoting an enhanced interaction between MAVS and UBC9. Furthermore, we demonstrated that this N-protein-induced change in MAVS SUMOylation leads to a reduction in its Lys63-linked ubiquitination, consequently inhibiting phosphorylation events involving IKKα, TBK1, and IRF3. As a result, the IFN-β signaling pathway is disrupted.

These findings elucidate the mechanism by which the N protein inhibits MAVS ubiquitination, resulting in the suppression of IFN-β expression and evasion of the host’s innate immune response. This knowledge can potentially contribute to the development of targeted therapeutic strategies to combat severe COVID-19 and enhance our overall understanding of the virus’s pathogenesis.

## 2. Materials and Methods 

### 2.1. Cells and Reagents

In this study, various cell lines, including HEK293T, VERO, VERO E6, Huh7, and HRT18, were obtained from the American Type Culture Collection (ATCC). HEK293T and Huh7 cells were cultured in DMEM media; VERO and VERO E6 cells were cultured in MEM media; and HRT18 cells were cultured in RPMI media. All media were supplemented with 10% fetal bovine serum and 100 U/mL penicillin/streptomycin. The cells were maintained at 37 °C in a humidified atmosphere with 5% CO_2_. It’s noteworthy that all cell lines used in the study were regularly tested and confirmed to be free from mycoplasma contamination. To trigger the innate immune response, the researchers utilized Sendai virus (SeV) stimulation.

### 2.2. Plasmid Constructs and Transfection

To generate the necessary plasmids for the study, the coding sequences of five human beta coronavirus N proteins (or truncated N proteins), UBC9 (or UBC9^C93A^ mutant), MAVS, SUMO3, and Ub-K63 (ubiquitin contains a substitution of arginine for all lysine residues, except the lysine at position 63) were chemically synthesized and amplified using standard PCR techniques. Subsequently, these amplified sequences were inserted into different vectors, namely pCDNA3.1, pET28a, pGEX-6P-1, pEGFP-N1, and pDsRED-mono-N1, as required for the subsequent experiments. To ensure the accuracy of the plasmid constructs, rigorous verification was conducted through DNA sequencing, which was performed by Wuhan Aoke Botai Biotechnology Co., Ltd. (Wuhan, China). For the transfection assay, the researchers followed the instructions provided by the manufacturer and utilized Lipofectamine™ 3000 Transfection Reagent (Cat No. L3000008) obtained from Thermo Fisher Scientific Inc. (Rockford, IL, USA).

### 2.3. Immunoblotting

For the immunoblot assay, whole cells were lysed with RIPA buffer (50 mM Tris, 150 mM NaCl, 1% NP-40, 0.5% sodium deoxycholate, 0.1% SDS, pH 7.4) supplemented with a protease inhibitor cocktail (containing 2 mM AEBSF, 0.3 μM Aprotinin, 130 μM Bestatin, 14 μM E64, and 10 μM Leupeptin) purchased from Beyotime Biotechnology. (P1005) (Shanghai, China) on ice for 30 min. The total concentration of protein was quantified using a BCA Protein Assay Kit (Beyotime, Cat No. P0010S) according to the manufacturer’s instructions. Subsequently, the protein was separated by SDS-PAGE and transferred to PVDF membranes (Millipore, MA, USA). After blocking with 5% skimmed milk, the membranes were incubated overnight at 4 °C with primary antibodies. Anti-FLAG (F1804) antibody was purchased from Sigma-Aldrich (St Louis, MO, USA). Anti-GFP (66002-1-Ig), anti-GST (66001-2-Ig), anti-UBC9 (60201-1-Ig) and anti-GAPDH (60004-1-Ig) antibodies were purchased from Proteintech Group (Wuhan, China). Anti-His_6_ (ab18184), anti-HA tag (ab13834), anti-K63 (ab179434), anti-SUMO2/3 (ab81371), anti-IKKα (ab32041), anti-p-IKKα at Ser176/180 (ab17943), anti-TBK1 (ab40676), anti-p-TBK1 at Ser172 (ab109272), anti-IRF3 (ab76409), anti-p-IRF3 at Ser396 (ab81371) antibodies were purchased from Abcam (Cambridge, MA, USA).

After three washes to remove any unbound primary antibodies, the membranes were probed with horseradish peroxidase-coupled (HRP) secondary antibodies. Subsequently, the membranes were exposed, and signals were recorded using Image Lab Software 5.1 (Bio-Rad, Hercules, CA, USA) following incubation with Clarity Western ECL Substrate (Bio-Rad, Cat No. 1705061). If necessary, the relative band intensity was quantified using ImageJ software (Version 1.8).

### 2.4. Coimmunoprecipitation (Co-IP)

Cells were harvested from 100-mm petri dishes and subjected to lysis using RIPA buffer (50 mM Tris, 150 mM NaCl, 1% NP-40, 0.5% sodium deoxycholate, 0.1% SDS, pH 7.4) supplemented with a protease inhibitor cocktail (containing 2 mM AEBSF, 0.3 μM Aprotinin, 130 μM Bestatin, 14 μM E64, and 10 μM Leupeptin) (Beyotime, Cat No. P1005) on ice for 30 min. Following lysis, immunoprecipitation was performed utilizing either an anti-Flag antibody (Sigma, Cat No. F1084) or an anti-His_6_ antibody (Abcam, Cat No. ab18184). Pierce™ Protein A/G Magnetic Beads (Thermo Scientific, Cat No. 88802) were employed for the immunoprecipitation. This procedure was conducted over a period of 3 h at 4 °C. The precipitated proteins underwent a series of five washes with PBST washing buffer (PBS with 0.1% Tween 20). Subsequently, these proteins were eluted from the beads using an elution buffer comprised of 50 mM sodium phosphate (pH 8.0), 300 mM sodium chloride, and 300 mM imidazole. The elution process was carried out at a temperature of 98 °C. The subsequent detection of the immunocomplexes was achieved through immunoblotting, employing specific antibodies corresponding to the targeted proteins.

### 2.5. Confocal Microscope

To ascertain the subcellular localization of UBC9 (or UBC9^C93A^ mutant) and N, a co-expression strategy was employed using pDsRED-mono-N1 (RFP Tag) containing the SARS-COV-2 N protein-coding genes and pEGFP-N1 (GFP Tag) containing the UBC9 or UBC9^C93A^ coding gene. These constructs were simultaneously introduced into HEK293T cells. Subsequently, the transfected cells were exposed to Sev for an additional 8 h, followed by PBS washing and fixation with 4% paraformaldehyde in PBS for a duration of 20 min. Endogenous UBC9 in HEK293T cells was also detected by FITC–conjugated Goat Anti-Mouse IgG(H + L) (Proteintech, Cat No. SA00003-1), which recognizes anti-UBC9 mouse antibody (Proteintech, Cat No. 60201-1-Ig). Specifically, HEK 293T cells were immobilized using 4% paraformaldehyde and permeabilized with Triton X-100 for 10 min. After washing with PBS and blocking with 5% skimmed milk, the cells were incubated with primary antibodies targeting UBC9 proteins overnight at 4 °C. After three washes with PBS, the cells were incubated with FITC-conjugated IgG for 1 h at room temperature. Prior to confocal microscopy PCM-2000 (NIKON, Melville, NY, USA) imaging, the cells were subjected to DAPI staining. 

### 2.6. SUMOylation and Ubiquitination Assay

To explore ubiquitination, cells were collected 24 h following transfection with the designated recombinant constructs, and they were subsequently exposed to SeV stimulation. Following this, cellular lysis was conducted using RIPA buffer (50 mM Tris, 150 mM NaCl, 1% NP-40, 0.5% sodium deoxycholate, 0.1% SDS, pH 7.4) supplemented with a protease inhibitor cocktail (containing 2 mM AEBSF, 0.3 μM Aprotinin, 130 μM Bestatin, 14 μM E64, and 10 μM Leupeptin) (Beyotime, Cat No. P1005) and 5 μM MG132 (Sigma-Aldrich, Cat No: M7449-200UL) on ice for 30 min. Following immunoprecipitation, the presence of ubiquitination was investigated utilizing the anti-ubiquitin (linkage-specific K63) antibody (Abcam, Cat No. ab179434). 

For the purpose of identifying SUMOylation, cells were subjected to lysis using RIPA buffer (50 mM Tris, 150 mM NaCl, 1% NP-40, 0.5% sodium deoxycholate, 0.1% SDS, pH 7.4) supplemented with a protease inhibitor cocktail (containing 2 mM AEBSF, 0.3 μM Aprotinin, 130 μM Bestatin, 14 μM E64, and 10 μM Leupeptin) (Beyotime, Cat No: P1005) and 20 μM N-Ethylmaleimide (Sigma-Aldrich, Cat No. E3876-5G) on ice for 30 min. After immunoprecipitation, the assessment of SUMOylation was performed using the anti-SUMO2/3 antibody (Abcam, Cat No. ab81371). 

### 2.7. Statistical Analyses

Statistical analyses were conducted using GraphPad Prism 5 (GraphPad Software, 5.0) and SPSS (v19.0). An unpaired Student’s *t*-test was utilized to compare the two groups, while an ANOVA was used for multiple group comparisons. The symbols *, **, ***, and “ns” were used to represent *p*-values of less than 0.05, less than 0.01, less than 0.001, and no significant difference, respectively.

## 3. Results

### 3.1. N Protein Interacts with Human SUMO-Conjugating E2 Enzyme UBC9

A previous study indicated that the N protein of SARS-CoV interacts with the human SUMO-conjugating E2 enzyme UBC9 and that mutations in UBC9 at the active site (specifically the C93A mutation) severely disrupt the interaction with the SARS-CoV N protein [10]. To investigate if the N protein of SARS-CoV-2 also interacts with UBC9, Co-IP experiments were performed. The results demonstrated that the N protein of SARS-CoV-2 specifically binds to UBC9 (Figure 1A). To further confirm this interaction, SARS-CoV-2 N protein and UBC9 (or UBC9^C93A^) were co-expressed in HEK293T cells and visualized using confocal microscopy. The results revealed that the N protein and UBC9 were co-localized in the cytoplasm. However, the UBC9^C93A^ mutation seems no longer present in the nucleus (Figure 1B, second lanes). We further evaluated the subcellular location of UBC9 and UBC9^C93A^ in HEK293T cells without N overexpression or Sev stimulation. The results showed that wild-type UBC9 is distributed in both the cytoplasm and nucleus, but the UBC9^C93A^ mutant is only distributed in the cytoplasm (Figure 1B, third and fourth lanes). Endogenous UBC9 in the HEK293T cell was also detected, and the results showed that endogenous UBC9 is distributed in both the cytoplasm and nucleus (Figure 1B, fifth lanes).

Subsequently, plasmids containing the N protein-coding genes with a fused HIS_6_ tag (in the pET28a vector) and the UBC9 or UBC9^C93A^ coding genes with a fused GST tag (in the pGEX-6P-1 vector) were co-expressed in BL21 (DE3) *E. coli* cells. After induction with IPTG (1 mM) for 6 h, cell lysates were subjected to Co-IP. The results confirmed that mutations in UBC9 at the C93A site impaired its ability to bind to the N protein (Figure 1C). We also verified the interaction of the SARS-COV-2 N protein with endogenous UBC9 in HEK293T cells. Co-IP assays proved that SARS-COV-2 N protein could bind endogenous UBC9 in HEK293T cells (Figure 1D).

Considering the conservation of the N protein among human infectious β coronaviruses [11], further investigation was conducted to determine if all five human β coronavirus (SARS-CoV, MERS-CoV, SARS-CoV-2, HCoV-OC43, HCoV-HKU1) N proteins could interact with UBC9. Co-expression of these N proteins in *E. coli* or HEK293T cells, followed by Co-IP experiments, demonstrated that all five human β coronavirus N proteins were capable of interacting with the SUMO-conjugating enzyme E2 UBC9 (Figure 2A,B).

Taken together, these results demonstrate that the human beta coronavirus N protein interacts with the human SUMO-conjugating E2 enzyme UBC9.

### 3.2. SARS-COV-2 N Protein Enhances Molecular Interaction between UBC9 and MAVS

Previous studies have reported that the N protein of SARS-CoV-2 interacts with MAVS and inhibits the Lys63-linked poly-ubiquitination of MAVS [9]. Based on our findings regarding the interaction between the N protein and UBC9, we further investigated whether the N protein mediates the molecular interaction between UBC9 and MAVS.

To examine this, we co-expressed the N protein, MAVS, and UBC9 in HEK293T cells and performed a Co-IP assay. The results revealed that MAVS could interact with UBC9 even in the absence of the N protein. However, the presence of the N protein enhanced the molecular interaction between UBC9 and MAVS (Figure 3A,B). 

It is known that the SARS-CoV-2 N protein consists of two domains, the N-terminal domain (NTD; 44–174) and the C-terminal domain (CTD; 255–364) [11] (Figure 3C). The NTD is responsible for RNA-binding, and the CTD for dimerization and oligomerization. Two domains are connected by a flexible linker, which contains Serine/Arginine-rich region and Leucine/Glutamine-rich region (LKR; 174–255). Here, we determined the domain of SARS-CoV-2 N protein involved in the interaction with UBC9 by constructing four mutants, N^44–419^, N^174–419^, N^255–419^, and N^364–419^, which lack N-arm, NTD, LKR, and CTD, in sequence, respectively. Co-IP results showed that, like full-length N protein, mutants N^44–419^ could interact with UBC9, except mutants N^174–419^, N^255–419^, and N^364–419^. This suggests that the NTD domain of the N protein interacts with the UBC9 protein (Figure 3C,D).

Taken together, these results demonstrate that the N-terminal domain (NTD) of the N protein interacts with the UBC9 protein, thereby enhancing the molecular interaction between UBC9 and MAVS.

### 3.3. The Interaction between MAVS and UBC9 Inhibits MAVS Ubiquitination by Enhancing Its SUMOylation

SUMOylation has a narrower range of enzymatic machinery compared to ubiquitylation. SUMOylation primarily relies on a single E1-activating enzyme complex called SAE1/SAE2 and a sole E2 SUMO-conjugating enzyme known as UBC9 [12]. Consequently, the increased molecular interaction observed between MAVS and UBC9 could potentially result in an up-regulation of SUMOylation on MAVS, impacting its function.

Previous studies have reported that, when the RIG-I/MAVS antiviral pathway is activated, polymeric chains of SUMO3, but not SUMO1 and SUMO2, are observed on MAVS [13]. Additionally, K63 ubiquitination is known to be essential for MAVS activation and propagation of the antiviral signaling cascade [14].

MAVS boasts several lysine residues distributed across different domains. For instance, previous research has reported that TRIM31 interacts with MAVS and orchestrates Lys63 (K63)-linked polyubiquitination on specific lysine residues, including Lys10, Lys311, and Lys461 [15]. Moreover, certain lysine residues, such as Lys461 and Lys500, have been identified as sites for SUMOylation on MAVS. Substituting either of these two sites with arginine, as demonstrated in cases like MAVS-K461R or MAVS-K500R, significantly inhibits poly-SUMOylation of MAVS. A double mutation of MAVS-K416RK500R not only hampers MAVS poly-SUMOylation but also drastically diminishes K63-linked MAVS polyubiquitination [16]. These findings strongly suggest that ubiquitination and SUMOylation of MAVS can indeed take place on the same lysine residue, indicating a close interaction between these two post-translational modification processes.

Given this interplay between ubiquitination and SUMOylation on the same lysine residue, we embarked on further investigations to ascertain whether the augmented interaction between MAVS and UBC9 curtails MAVS ubiquitination by intensifying its SUMOylation. First, we examined the endogenous expression level of UBC9 in different cell lines (HEK293T, Vero, Vero E6, Huh7, and HRT18). The results from Western blotting showed that UBC9 is highly expressed in HEK293T cells but underexpressed in Vero E6 cells (Figure 4).

Next, we co-expressed Ub-K63-his_6_ (ubiquitin containing a substitution of arginine for all lysine residues, except the lysine at position 63), SUMO3-HA, MAVS-Flag, and N-GFP in HEK293T cells or Vero E6 cells. Subsequently, we conducted Co-IP analysis employing an anti-Flag antibody to examine alterations in the ubiquitination and SUMOylation of MAVS.

In HEK293T cells (endogenous UBC9 is highly expressed), we made an interesting observation. As we incrementally expressed the N protein, there was a notable decrease in K63 ubiquitination and a simultaneous increase in SUMOylation of MAVS (Figure 5A).

However, the pattern differed in Vero E6 cells, where endogenous UBC9 is underexpressed. In this context, we did not observe the same trend. There was no significant decrease in K63 ubiquitination, nor was there an increase in SUMOylation of MAVS when we incrementally expressed the N protein, as shown in Figure 5B.

Taken together, these results demonstrate that the N protein interacts with the UBC9 protein, enhancing the molecular interaction between UBC9 and MAVS. This enhanced interaction between MAVS and UBC9 inhibits MAVS ubiquitination by enhancing SUMOylation.

### 3.4. The Interaction of N Protein and UBC9 Plays a Crucial Role in Regulating the SUMOylation and Ubiquitination Modifications of MAVS

To further investigate this phenomenon, we conducted experiments in HEK293T cells, where endogenous UBC9 is highly expressed, using a lysine-deficient SUMO3^KR^ mutant (a SUMO3 mutant with arginine substitutions for all lysine residues). In these cells, we co-expressed K63-his_6_, WT SUMO3, or SUMO3^KR^ mutants, along with MAVS-Flag and N-GFP. Surprisingly, in contrast to the significant decrease in K63 ubiquitination and increase in SUMOylation observed when overexpressing SUMO3, there was no such effect when overexpressing the SUMO3^KR^ mutant in HEK293 cells (Figure 6A). Previous studies have indicated that lysine-deficient SUMO2 exhibits similarities to wild-type SUMO2, except for SUMO polymerization [17]. In light of this report, our discovery suggests that the polymerization of SUMO3 could potentially impact the K63 ubiquitination of MAVS.

As we have demonstrated that the UBC9^C93A^ mutant impairs its ability to bind to the N protein (Figure 1C), we further explored its impact. We co-expressed K63-his_6_, WT UBC9, or the UBC9^C93A^ mutant, along with MAVS-Flag and N-GFP in Vero E6 cells, where endogenous UBC9 is underexpressed. As expected, the overexpression of the UBC9^C93A^ mutant did not lead to a significant decrease in K63 ubiquitination or an increase in SUMOylation of MAVS in Vero E6 cells (Figure 6B).

As the UBC9^C93A^ mutant is a catalytically inactive mutant of UBC9, which is not able to bind SUMO, the decrease in SUMOylation of MAVS in Vero E6 cells may also be influenced by this factor. We further verified if the interaction of UBC9 and N inhibits MAVS ubiquitination by enhancing its SUMOylation by using N-truncated mutants. Our previous results revealed that UBC9 interacts with the NTD domain of the SARS-CoV-2 N protein (Figure 3C,D). In order to explore the impact of this interaction, we co-expressed WT N protein, the N^44–419^ mutant (containing the NTD domain), or the N^174–419^ mutant (lacking the NTD domain), with K63-his_6_, SUMO3-HA, and MAVS-Flag in HEK293T cells. The results demonstrated that both the WT N protein and the N_44-419_ mutant (with the NTD domain) caused a notable decrease in K63 ubiquitination and a simultaneous increase in SUMOylation of MAVS. However, the N^174–419^ mutant, lacking the NTD domain and failing to interact with endogenous UBC9, did not result in a significant decrease in K63 ubiquitination or an increase in SUMOylation of MAVS in HEK293T cells (Figure 6C).

Taken together, these results suggest that the interaction of N protein and UBC9 plays a crucial role in regulating the SUMOylation and ubiquitination modifications of MAVS during virus infection. 

### 3.5. UBC9 Plays a Critical Role in the Process of Impaired IFN I Response Caused by the Interaction between the N Protein and MAVS during Virus Infection

The reduction in phosphorylation of IKKα, TBK1, and IRF3 indicates a compromised IFN I response in the presence of the SARS-CoV-2 N protein [18]. Additionally, our study delved into the impact of incrementally increasing N protein expression on the phosphorylation levels of IKKα, TBK1, and IRF3 in HEK293T cells, known for their high expression of UBC9. The results demonstrated a significant reduction in phosphorylation levels in HEK293T cells (Figure 7A,C). Conversely, this phenomenon was less pronounced in UBC9-underexpressed VeroE6 cells (Figure 7B,D).

These results demonstrate that UBC9 plays a critical role in the impaired IFN I response caused by the interaction between the N protein and MAVS during virus infection.

## 4. Discussion

The ongoing global pandemic caused by the severe acute respiratory syndrome coronavirus 2 (SARS-CoV-2) has underscored the urgent need for a comprehensive understanding of the virus and its interactions with host cellular machinery. Despite extensive efforts to combat the virus, the molecular mechanisms by which SARS-CoV-2 overcomes host defense systems to cause respiratory symptoms and potentially fatal outcomes remain largely elusive [19]. 

Host defense responses are initiated by the recognition of pathogen-related molecular patterns through various receptors [20,21]. Upon sensing viral RNA duplexes, key players like RIG-I and Toll-like receptor family members trigger a signaling cascade that induces the transcription of interferons (IFN-β and IFN-α) to mount an antiviral defense [14,22,23,24,25]. In this process, Lys63-linked polyubiquitination of MAVS plays a crucial role in facilitating the aggregation of MAVS, thereby initiating IFN-β signaling and downstream antiviral responses [9,14,26]. It has been demonstrated that if the SARS-CoV-2 N protein inhibits Lys63-linked polyubiquitination and aggregation of MAVS, then the innate antiviral immune response initiated by MAVS is inhibited [9]. However, the precise molecular mechanism by which the SARS-CoV-2 N protein regulates the polyubiquitination of MAVS remains elusive.

In this study, we initially found that there is an interaction between the human beta coronavirus N protein and the SUMO-conjugating E2 enzyme UBC9, as illustrated in Figure 1 and Figure 2. This interaction holds substantial significance, particularly in light of previous reports indicating that the SARS-CoV-2 N protein plays a pivotal role in modulating MAVS’s interaction with RIG-I and TRIM31, thereby suppressing innate immune signaling at the MAVS level [9]. Our subsequent investigations have provided further validation that the interaction between N and UBC9 amplifies the molecular interplay between UBC9 and MAVS (Figure 3A,B). In addition, we have identified that the N protein’s interaction with UBC9 is contingent on the presence of the N-terminal domain (NTD) (Figure 3C,D). 

In addition to its role in ubiquitination, SUMOylation is a crucial protein modification process with multifaceted involvement in antiviral defense mechanisms, as supported by recent research [12,14,27]. Notably, recent studies have illuminated the ability of the SARS-CoV-2 Nsp5 protein to induce MAVS SUMOylation through its interaction with UBC9 [28]. Building upon our initial findings, our subsequent study aimed to delve deeper into the potential implications of the intensified molecular interaction between MAVS and UBC9 on both MAVS ubiquitination and SUMOylation. To explore this further, we conducted experiments using cell models with varying UBC9 expression levels and observed that UBC9 is highly expressed in HEK293T cells but underexpressed in Vero E6 cells (Figure 4). Intriguingly, we discovered that the SARS-CoV-2 N protein inhibits MAVS ubiquitination while enhancing its SUMOylation in HEK293T cells but not in VERO E6 cells (Figure 5A,B). Notably, this phenomenon indicates that the N protein-induced SUMOylation of MAVS is contingent upon the presence of UBC9.

In the subsequent experiments, we proceeded to overexpress the SUMO3^KR^ and UBC9^C93A^ mutants in HEK293T and VERO E6 cells, respectively. Interestingly, we observed that the lack of SUMO3 polymerization or the absence of interaction between N and UBC9 both resulted in the inhibition of N protein-induced MAVS SUMOylation (Figure 6A,B). Furthermore, our previous findings indicated that, in the absence of the NTD domain, the N protein fails to interact with UBC9. This lack of interaction also results in an inability to inhibit MAVS ubiquitination while simultaneously enhancing its SUMOylation (Figure 6C).

Hence, we have successfully elucidated how the SARS-CoV-2 N protein enhances the molecular interaction between UBC9 and MAVS. This interaction, as we have shown, hinders MAVS ubiquitination by promoting its SUMOylation. As a result, we proceeded to investigate whether this interaction has any impact on the MAVS-TBK1-IKK complex within the MAVS-IRF3 signaling cascade. As anticipated, the gradual increase in N protein expression led to reduced phosphorylation levels of IKKα, TBK1, and IRF3 in UBC9 highly expressed HEK293T cells, while this effect was not observed in UBC9 underexpressed VERO E6 cells (Figure 7).

In summary, our results highlight the significance of the interaction between the SARS-CoV-2 N protein and UBC9. This interaction appears to hinder the ubiquitination of MAVS by facilitating its SUMOylation, subsequently impairing the host’s interferon (IFN) type I response. These findings illuminate the complex interplay between various post-translational modifications and provide insights into the regulatory mechanisms that govern MAVS-mediated antiviral responses.

## Figures and Tables

**Figure 1 viruses-15-02304-f001:**
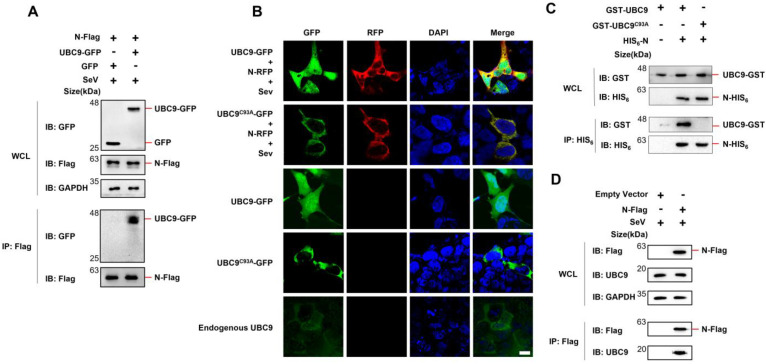
SARS-COV-2 N protein interacts with UBC9. (**A**) pCDNA3.1 (3XFlag Tag) with N protein-coding genes and pEGFP-N1 (GFP Tag) with UBC9-coding genes were co-expressed in HEK293T cells. After a 24-h lipofectamine transfection followed by 8 h of SeV stimulation, cell lysates were collected for Co-IP assays. (**B**) pDsRED-mono-N1 (RFP Tag) with SARS-COV-2 N protein-coding genes and pEGFP-N1 (GFP Tag) with UBC9 or UBC9^C93A^ coding genes were expressed or co-expressed in HEK293T cells. After a 24-h lipofectamine transfection followed by 8 h of SeV stimulation, cells were stained with DAPI before confocal microscopy imaging. Endogenous UBC9 in HEK293T cells was also detected by FITC-conjugated Goat Anti-Mouse IgG(H + L), which recognizes anti-UBC9 mouse antibodies. Scale bars: 10 μm. (**C**) pET28a (HIS_6_ tag, Kan^+^) with SARS-COV-2 N protein coding genes and pGEX-6P-1 (GST tag, Amp^+^) with UBC9 or UBC9^C93A^ coding genes were co-expressed in BL21 (DE3) *E. coli*. cell lysates were subjected to Co-IP after IPTG (1 mM) induction for 6 h. (**D**) pCDNA3.1 (3XFlag Tag) with SARS-COV-2 N protein-coding genes was over-expressed in HEK293T cells. After a 24-h lipofectamine transfection followed by 8 h of SeV stimulation, cell lysates were collected for Co-IP assays.

**Figure 2 viruses-15-02304-f002:**
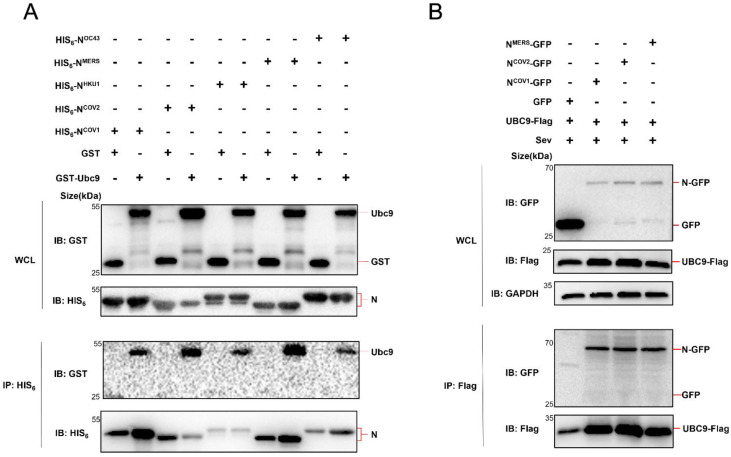
Human beta coronavirus N protein interacts with UBC9. (**A**) pET28a-N (HIS_6_ tag, Kan^+^) containing five human beta coronavirus N protein-coding genes and pGEX-UBC9 (GST tag, Amp^+^) harboring the UBC9 coding gene were co-expressed in BL21 (DE3) *E. coli* Cell lysates were prepared and subjected to Co-IP following IPTG induction (1 mM) for 6 h. (**B**) pEGFP-N1 (GFP Tag) with three human beta coronavirus N protein-coding genes and pCDNA3.1 (3XFlag Tag) with the UBC9 coding gene were co-expressed in HEK293T cells. After a 24- h lipofectamine transfection followed by 8 h of SeV stimulation, cell lysates were collected for Co-IP assays.

**Figure 3 viruses-15-02304-f003:**
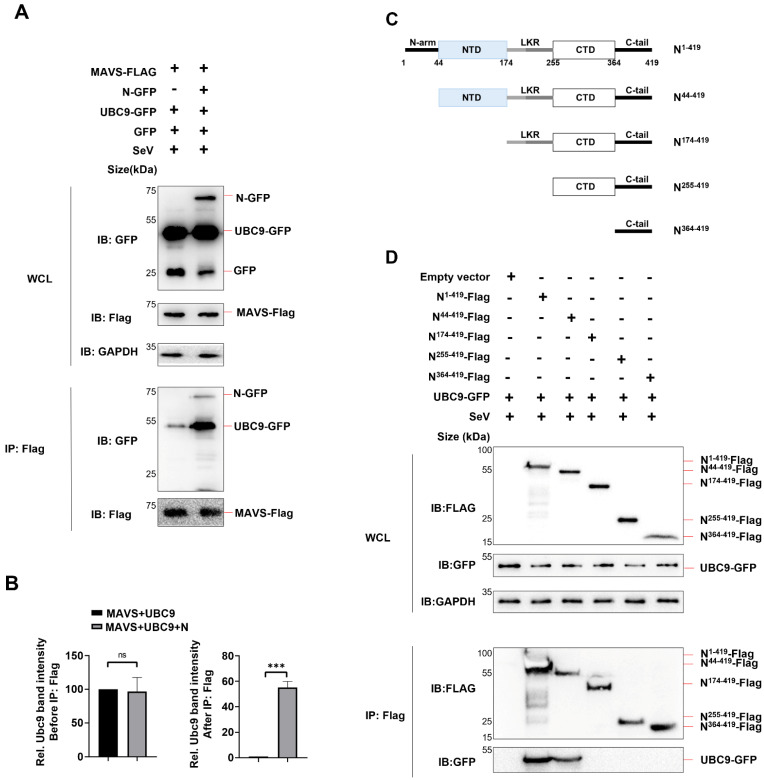
SARS-COV-2 N protein enhanced the molecular interaction between UBC9 and MAVS. (**A**) pEGFP-N1 (GFP Tag) with SARS-COV-2 N protein or UBC9 coding genes and pCDNA3.1 (3XFlag Tag) with MAVS coding gene were co-expressed in HEK293T cells. After a 24-h lipofectamine transfection followed by 8 h of SeV stimulation, cell lysates were collected for Co-IP assays. (**B**) Relative Ubc9 band intensity before or after IP: Flag in (**A**) were calculated by using ImageJ in triplet replicates. ns: no significant difference, *** *p* < 0.001. (**C**) Schematic diagram of wild-type SARS-CoV-2 N protein and truncated mutants. (**D**) pEGFP-N1 (GFP Tag) with UBC9 coding genes and pCDNA3.1 (3XFlag Tag) with SARS-COV-2 N protein or truncation coding gene were co-expressed in HEK293T cells. After a 24-h lipofectamine transfection followed by 8 h of SeV stimulation, cell lysates were collected for Co-IP assays.

**Figure 4 viruses-15-02304-f004:**
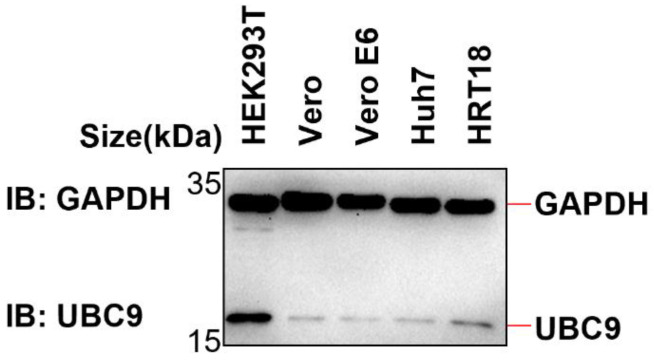
The expression of UBC9 in HEK293T, Vero, Vero E6, Huh7, and HRT18 cells was detected by Western blotting.

**Figure 5 viruses-15-02304-f005:**
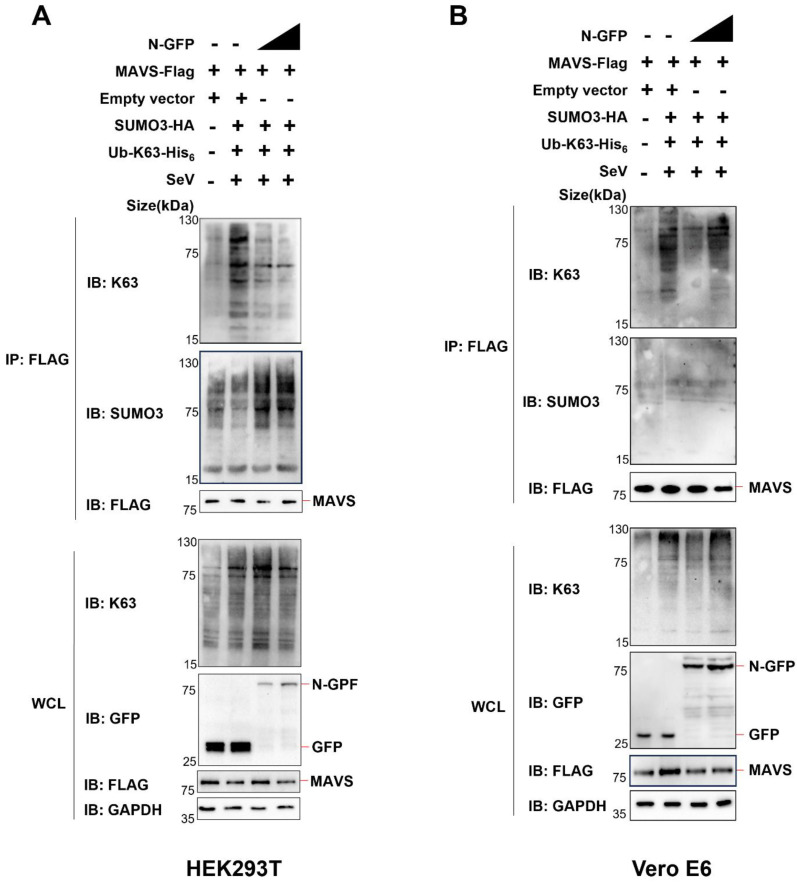
The interaction between the SARS-CoV-2 N protein and UBC9 inhibits MAVS ubiquitination by enhancing its SUMOylation. (**A**,**B**) pEGFP-N1 (GFP Tag) with SARS-COV-2 N coding genes or pCDNA3.1 with MAVS-Flag, SUMO3-HA, or K63-His6 coding genes were co-expressed in HEK293T cells (endogenous UBC9 highly expressed) or Vero E6 cells (endogenous UBC9 underexpressed). After a 24-h lipofectamine transfection followed by 8 h of SeV stimulation, cell lysates were collected for Co-IP assays.

**Figure 6 viruses-15-02304-f006:**
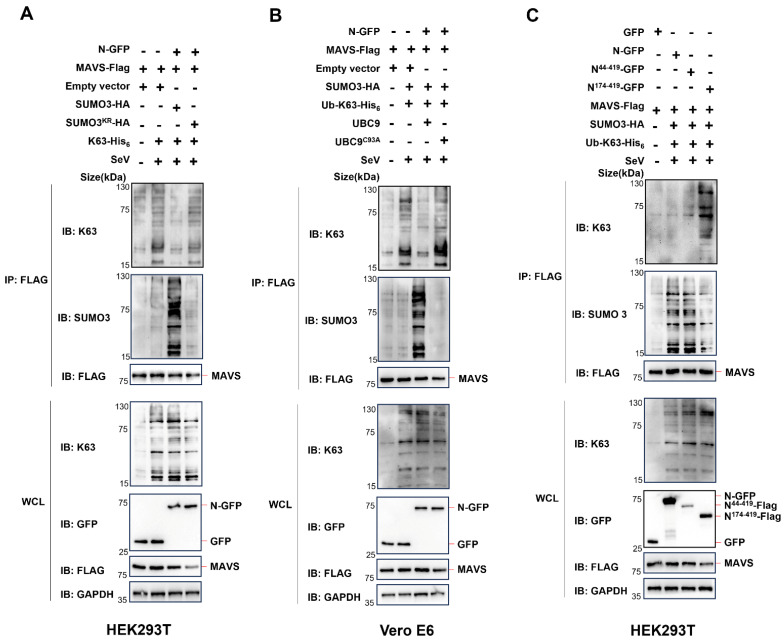
The interaction of N protein and UBC9 plays a crucial role in regulating the SUMOylation and ubiquitination modifications of MAVS. (**A**) pEGFP-N1 (GFP Tag) with SARS-COV-2 N coding genes or pCDNA3.1 with MAVS-Flag, SUMO3-HA or SUMO3^KR^-HA, K63-His_6_ coding genes were co-expressed in HEK293T cells (endogenous UBC9 was highly expressed). After a 24-h lipofectamine transfection followed by 8 h of SeV stimulation, cell lysates were collected for Co-IP assays. (**B**) pEGFP-N1 (GFP Tag) with SARS-COV-2 N coding genes or pCDNA3.1 with MAVS-Flag, SUMO3-HA, UBC9 or UBC9^C93A^, K63-His_6_ coding genes were co-expressed in Vero E6 cells (endogenous UBC9 underexpressed). After a 24-h lipofectamine transfection followed by 8 h of SeV stimulation, cell lysates were collected for Co-IP assays. (**C**) pEGFP-N1 (GFP Tag) with SARS-COV-2 N, N^44–419^ or N^174–419^ coding genes or pCDNA3.1 with MAVS-Flag, SUMO3-HA, and K63-His_6_ coding genes were co-expressed in HEK293T cells (endogenous UBC9 highly expressed). After a 24-h lipofectamine transfection followed by 8 h of SeV stimulation, cell lysates were collected for Co-IP assays.

**Figure 7 viruses-15-02304-f007:**
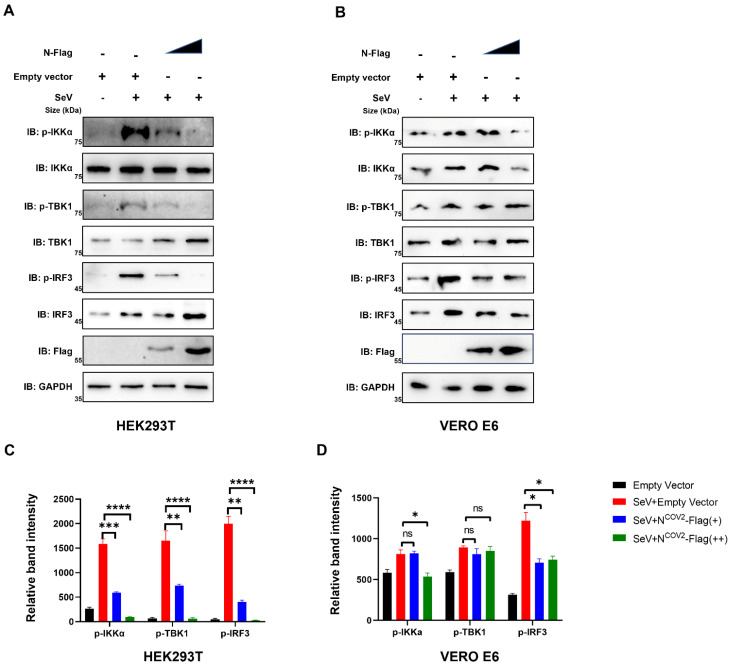
UBC9 plays a critical role in the process of impaired IFN I response caused by the in-teraction between the N protein and MAVS during virus infection. (**A**,**B**) pCDNA3.1 with SARS-CoV-2 N protein-coding genes was gradually increased in HEK293T cells or VERO E6 cells. After a 24-h lipofectamine transfection followed by 8 h of SeV stimulation, cell lysates were collected for Western blotting. (**C**,**D**) The relative band intensity of Western blotting results (a relative band quantification of phosphorylation protein to total protein) in (**A**,**B**) were calculated by using ImageJ in triplet replicates. ns: no significant difference, *p* > 0.05; *: *p* < 0.05; **: *p* < 0.01; ***: *p* < 0.001; ****: *p* < 0.0001.

## Data Availability

All processed data are included in the manuscript.

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
