# Peer review of "The Interaction between SARS-CoV-2 Nucleocapsid Protein and UBC9 Inhibits MAVS Ubiquitination by Enhancing Its SUMOylation"

_viruses, 2023, doi:10.3390/v15122304_

Round 1

Reviewer 1 Report (Previous Reviewer 2)

Comments and Suggestions for Authors

 The revised manuscript addressed some of my concerns; however, it's crucial to underline that MG132's role is to block proteasome activity within cells; hence, it should be applied prior to cell collection. Post-harvest, protease inhibitors such as PMSF and leupeptin must be used to prevent protein degradation. Unless there is compelling evidence to support their chosen methodology, I recommend that the experiments be re-conducted. Additionally, the manuscript lacks clarity regarding whether protein denaturation was conducted before the ubiquitination and SUMOylation assays to preclude the detection of these modifications on proteins interacting with MAVS. 

Lysine-deficient SUMO2/3 behaves very similarly to wild-type SUMO2/3, except for SUMO polymerization (PMID: 25218447, 28112733, 37188808). This implies that MAVS is still subject to SUMOylation upon overexpression of the SUMO3KR mutant in HEK293 cells. Therefore, it is necessary for the authors to provide additional validation and explanation for the results presented in Figure 7a. 

The authors did not employ actual SARS-CoV-2 to infect the cells; therefore, descriptions such as "during SARS-CoV-2 infection" are misleading and should be revised.

Author Response

Reviewer 2 Report (New Reviewer)

Comments and Suggestions for Authors

In severe cases of Covid-19, it has been observed that the IFN-1 response is significantly impaired, resulting in an increased viral load.  The viral nucleocapsid protein has been implicated in this impaired immune response, though the details of the mechanism by which this manipulation of it by N has not been completely elucidated.  In this excellent manuscript, the mechanism by which N promotes the evasion of the immune response is investigated.  The N protein has previously been shown to interact with the RIG-I protein, resulting in an inhibition of IFN-1 signaling.  In this study, the characterization of the role of the N protein in this process is taken to the next level.  Based on exquisitely designed experimental approaches and totally convincing data, the study firmly and convincingly shows that the SARS-CoV-2 N protein inhibits the IFN-1 response by binding to the UBC9 protein.  This interaction prevents the ubiquitination of MAVS by virtue of stimulating its SUMOylation, which, in turn, inhibits the IFN-1 response.  This is considered an outstanding manuscript that significantly advances our understanding of the interactions of the SARS-CoV-2 N protein that facilitate its impairment of the IFN response to the virus, in, turn, promoting the infection.

Comments on the Quality of English Language

Only minor corrections required.

Author Response

Reviewer 3 Report (New Reviewer)

Comments and Suggestions for Authors

Review report on manuscript 2704677 submitted to Viruses 

The interaction between SARS-CoV-2 Nucleocapsid protein and UBC9 inhibits MAVS ubiquitination by enhancing its SUMOylation

Congcong Huang, Yiping Yin, Pan Pan, Yanping Huang, Siwei Chen, Junkai Chen, Ju Wang, Guoqing Xu, Xuan Tao, Xiao Xiao, Jian Li, Jing Yang, Zhixiong Jin, Bei Li, Zhaohui Tong, Weixing Du, Long Liu and Zhixin Liu 

Huang et al. characterize in this article the functional interaction between SARS-CoV-2 N and the SUMO conjugating enzyme UBC9. They demonstrate that, through the interaction with N, UBC9 enhances MAVS SUMOylation leading its reduced ubiquitination. This finally leads to a reduced activation of the transcription factor IRF3, responsible of the induction of IFN-β upon viral infection.

This study is interesting and quite comprehensive in the different experiments performed. Nevertheless, I have a few major concerns, listed below. 

1.   Subcellular localization of UBC9-GFP and N-RFP (Figure 1B). The pictures presented are of poor quality, particularly concerning DAPI. The authors must reproduce the experiment and present pcitures of better quality, with several cells on a field. It would be interesting to have cells expressing UBC9-GFP without N-RFP, in order to see if N overexpression modifies UBC9 subcellular localization (both wt and C93A mutant). The authors should also take pictures (at least present them in supplementary) of cells that have not been “activated” (i.e. infected by SeV). Since the authors show that UBC9 is highly expressed in HEK293T, and use an antibody against UBC9, why did they not performed the subcelular localization of endogenous UBC9 ? Moreover I disagree with authors conclusions : UBC9 C93A does not disrupt the subellular colocalization of both proteins. This UBC9 mutant is no more localized to nucleus, but only in the cytoplasm, just like N-RFP.

2.   Determination of N domains required for UBC9 interaction (Figure 4D). The authors performed N-term successive truncation mutants of N, and showed that if the NTD domain is removed, UBC9 does not interact anymore with N. They authors then conclude that the NTD of N for UBC9 interaction. In order to aqscertain this, the authors should perform a co-ip with NTD (aa 44-174 only) and UBC9. 

3.   UBC9 plays a critical role in IFN-I response (figure 8). The blots presented in figure 8B are not beautifull (and quite different to those of a previous version of this manuscript). These western-blots have been performed several times, since the authors present a densitometric analysis of the bands, but they do not tell how many replicates were performed. Furthermore, the authors do not explain how they proceeded for the band quantification. I guess it is a relative band quantification of P-protein to total protein (e.g. p-IKKα relative to total IKKα) ? Since MAVS induces the phosphorylation of the IRF3 transcription but also the activation of the NF-κB transcription factor (through phosphorylation and subsequent ubiquitination and degradation of IκB), the authors could also perform western-blots of IκB to analyse if NF-κB activation is also regulated through N-UBC9 interaction. Moreover, it would be nice to demonstrate IFN-β induction by performing a RTqPCR analysis of its transcript.

Minor comments

1.     In the introduction, the authors have to be more precise in the description of the innate immune pathway. RIG-I specifically recongizes double-stranded RNA as well as tri-P 5’-RNA; RIG-I is ubiquitinated by TRIM25 and RIPLET; upon RIG-I interaction, MAVS is ubiquitinated by TRIM31.

2.     In the material and methods (plasmids), the authors performed PCR to amplifiy the coding DNA sequence of the different proteins of interest in order to generate different plasmid constructs. The authors should cite the source of DNA.

3.     In the material and methods (co-immunoprecipitation), the authors have to give composition of washing buffer.

4.     In the material and methods (SUMOylation and ubiquitination assays), the authors have to give composition of lysis buffer.

5.     For most of experiments presented, the authors activate the innate immune pathway by infecting cells with SeV for 8h prior cell lysis, etc. What is happening if cells are not activated ? The authors should present at least the experiments of non-activated cells, in supplementary information (figs 1A, 1B, 1D, 4A, 4D, 6A, 6B, 8A, 8B).

6.     The authors use UBC9 C93A mutant, based on a previous study showing that it disrupts UBC9-N interaction. This UBC9 mutant is no more able to bind SUMO, and thus is a “catalytically” inactive mutant (for example, DOI: 10.2337/db06-1100). The authors should discuss this point.

7.     I suggest to combine figures 2 and 3

8.     Figure 4C. How many times the co-ip has been performed to generate this quantification ? I understand how we can measure the relative abundance of UBC9-GFP on WCL (relative to GAPDH), but on the IP how was performed the relative quantification ? Furthermore the quantification after IP shows that UBC9 is not interacting with MAVS in absence of N, but it is not the case. We can observe a clear band on the co-IP. I suggest to remove these quantifications that are misleading and unnecessary.

9.     Lane 255 ref.12 is incorrect. It is ref.11 that has to be cited.

10.  Figure 6 : we can clearly observe a difference without and with N expression upon MAVS ubiquitination and SUMOylation. But I am less enthusiastic when the authors claim that when N is more expressed, there is a notable decrease in MAVS ubiquitination and a simultaneous increase in its SUMOylation. It would be nice to have a blot showing the abundance of UBC9 in the different conditions, is it known if N can induce UBC9 expression ?

11.  Lane 200 a word is missing (…) N protein could bind endogenous UBC9 (…)

12.  Lanes 372-375 : rephrase, almost duplicated sentences.

13.  Lances 380-382 : rephrase, the sentence is not clear.

Figure 8 Cand D : portein band intensity => protein band intensity (and rather relative level of phospho proteins).

Comments on the Quality of English Language

I detected some typing errors that I mentioned in my comments to the authors. 

I suggest having the manuscript proofread by someone fluent in English in order to improve the quality of the text.

Round 2

Reviewer 1 Report (Previous Reviewer 2)

Comments and Suggestions for Authors

I have no further comments.

Author Response

Dear Reviewer,

We would like to express our sincere gratitude for your meticulous and thorough review of my manuscript. Your attention to detail and insightful feedback have been invaluable in enhancing the quality and clarity of the content. Your dedication to the peer review process has significantly contributed to the overall improvement of this work. We truly appreciate the time and effort you have devoted to ensuring the rigor and accuracy of the research. 

Best regards,
Zhixin Liu

Reviewer 3 Report (New Reviewer)

Comments and Suggestions for Authors

Comments and Suggestions for Authors
(will be shown to authors)

Review report on manuscript 2704677 submitted to Viruses 

The interaction between SARS-CoV-2 Nucleocapsid protein and UBC9 inhibits MAVS ubiquitination by enhancing its SUMOylation

Congcong Huang, Yiping Yin, Pan Pan, Yanping Huang, Siwei Chen, Junkai Chen, Ju Wang, Guoqing Xu, Xuan Tao, Xiao Xiao, Jian Li, Jing Yang, Zhixiong Jin, Bei Li, Zhaohui Tong, Weixing Du, Long Liu and Zhixin Liu 

Huang et al. characterize in this article the functional interaction between SARS-CoV-2 N and the SUMO conjugating enzyme UBC9. They demonstrate that, through the interaction with N, UBC9 enhances MAVS SUMOylation leading its reduced ubiquitination. This finally leads to a reduced activation of the transcription factor IRF3, responsible of the induction of IFN-β upon viral infection.

The manuscript has been improved and my major comments have been suitably addressed, I wish to thank the authors for the modifications of Figure 1B, that clearly show the subcellular localization of UBC9-GFP in presence or absence of N-RFP, and in “mock” condition (i.e. in cells that have not been activated by SeV infection).

Concerning my minor comments, most have been addressed.

Comment 9 (reviewing_V1)

To my opinion, ref #12 is not correct in the sentence at line 258-259. It is ref #11 that has to be cited. I performed screenshots, for better understanding.

Sentence at line 258-259:

Ref #11 and #12:

So the reference related to the domain composition of N is Masuo T (Ref #11) and not Fan Y et al. (Ref #12)

Comment 11 (reviewing_V1)

In sentence now at lane 219-220, a word is missing. I performed a screenshot, for better understanding.

I suggest to add (in bold red) : Co-IP assays proved that SARS-CoV-2 N protein could bind endogenous UBC9 in HEK293T cells (figure 1D).

Comment 12 (reviewing_V1)

There are two sentences at lanes 376-379, that are very similar, and I suggest the authors to rephrase it. I performed a screenshot, for better understanding.

Comments on the Quality of English Language

The manuscript has been carefully revised by a native speaker.

Author Response

Dear Reviewer,

We extend our sincere appreciation for your efforts in reviewing our manuscript. Your comments have been addressed and highlighted in the new version of the manuscript. We are truly grateful for the time and dedication you have invested in evaluating and improving the quality of the manuscript. Your expertise and valuable insights have significantly contributed to enhancing the overall clarity and merit of the work. Thank you once again for your time, effort, and commitment to the peer review process.

Best regards, 

Zhixin Liu

This manuscript is a resubmission of an earlier submission. The following is a list of the peer review reports and author responses from that submission.

Round 1

Reviewer 1 Report

Comments and Suggestions for Authors

This manuscript reports that SARS-CoV2 N protein interacts with UBC9 and proposes this interaction increases the SUMOylation of MAVS thereby decreasing MAVS ubiquitination and decreasing the interferon response.  Since it has already been shown the SARS-CoV N protein interacts with UBC9 and that SARS-CoV2 N decreases MAVS ubiquitination and the interferon response, the proposed advancement here is showing that UBC9-dependent SUMOylation is the mechanism that prevents MAVS ubiquitination and the interferon response.  But what is shown here is only a correlation, not a cause-and-effect relationship, making the manuscript, as presented only a minimal incremental advance in knowledge. 

The authors are encouraged to design and perform experiments that directly link SUMOylation to the effects on ubiquitination and interferon responses.  Options include testing an N mutant that fails to bind UBC9; in UBC9-low cells (e.g., VeroE6), overexpressing WT or catalytically mutant UBC9; using non-conjugatable SUMO3; or some other experiment that goes beyond correlation to show a cause and effect.

Minor comments:

Figure 1B – colocalization does not confirm an interaction

Figures 2 and 3 – demonstrating interaction of N with endogenous UBC9 would improve the impact of the manuscript.  Surely there is enough UBC9 in HEK293T cells to demonstrate an interaction.  Although more challenging, showing an interaction in a relevant cell type and during an infection would be most impactful.

Figure 6 – the authors need to describe what the “K63” gene is.  Is this ubiquitin (WT or mutant)?

Figure 6 – this is the main point of the paper and is not very convincing.  Another control would be helpful.  For example, co-transfection with a non-conjugatable SUMO3 allele should not increase MAVS SUMOylation and therefore should not decrease MAVS ubiquitination.  Some type of mutant analysis is necessary to provide more compelling data.

Figure 6 and Figure 7 – these figures disappointingly use a different set of cells.  The experiment in Figure 6 (and Figure 5) should be performed in A549 cells and the experiment in Figure 7 should be performed in VeroE6 cells to strengthen the correlation between UBC9 levels and the observed effects.

Comments on the Quality of English Language

Acceptable

Reviewer 2 Report

Comments and Suggestions for Authors

In this manuscript, the authors tried to demonstrate that the SARS-CoV-2 N protein inhibits MAVS K63-linked ubiquitination by enhancing the interaction between MAVS and UBC9. However, the manuscript exhibits significant flaws and requires supplementary experiments to strengthen its findings

Major points:

1 The primary concern pertains to the ubiquitination and SUMOylation assay. Firstly, the presence of MG132 inhibits intracellular proteasome activity, thereby enabling the accumulation of ubiquitinated and/or SUMOylated proteins. Curiously, the authors assert that they introduced MG132 into the lysis buffer (line 143-144). Furthermore, all of these assays in this manuscript lack pulldown control bands. The author should incorporate pulldown control bands to validate that equivalent protein quantities were pulled down from different samples. Lastly, preceding research has demonstrated that the N protein undergoes SUMOylation (PMID: 37515286, 15848177, 16998888, 17037517), and that SUMOylation of MAVS enhances MAVS activity (PMID: 31806367, 37188808, 34858407). The authors omitted the denaturation of samples, which could disrupt non-covalent interactions, during the ubiquitination and SUMOylation assay. This may potentially account for the observed contradictory trend in MAVS SUMOylation compared to prior literature.

2 The authors have not provided evidence for “ubiquitination and SUMOylation occur on the same lysine residue”. Moreover, it should be noted that SUMO and ubiquitin are capable of forming hybrid chains (PMID: 25218447, 37188808). It is recommended that the authors address these aspects with clarity in their writing.

3 Merely replicating the findings of the previous article in Fig 4b adds little value to this manuscript. Instead, the authors should identify the specific domains where MAVS and N protein bind to UBC9. Furthermore, it is important to illustrate whether and how the N protein facilitates the direct interaction between UBC9 and MAVS.

4 Numerous articles have already documented the inhibitory effect of the N protein on the IFN-I signaling pathway and its potential to enhance the inflammation pathway. Therefore, I am puzzled as to why Fig 7 replicates experiments related to the function of the N protein. Additionally, it's worth noting that the observed trend of IKKa in this figure diverges from what was presented in the previous article (PMID: 33895773). Moreover, considering the entirety of the manuscript, it would be valuable for the authors to explore the impact of UBC9 on IFN-I activity and MAVS function (like aggregation).

Specific points:

Fig 1. What was the duration of cellular stimulation by SeV? Moreover, what accounts for the presence of SeV infection in the cells depicted in Fig 1A, while the cells in Fig 1B appear to be devoid of SeV infection?

Fig 1C and Fig 2. The molecular weight of Ubc9 protein is smaller than that of GST. To enhance the robustness of the findings, it is recommended to either excise UBC9 from GST or employ alternative tags such as Strep.

Comments on the Quality of English Language

Requires extensive English language editing